# Effect of proprioceptive neuromuscular facilitation on patients with chronic ankle instability: A systematic review and meta-analysis

Yikun Yin[1], Jialin Wang[2]*, Qihan Lin[3], Yinghang Luo[4], Yongsheng Liu[4], Junzhi Sun[2]

1 School of Sport Human Science, Beijing Sport University, Beijing, China, 2 Institute of Sports Medicine and Health, Chengdu Sport University, Chengdu, China, 3 College of Physical Education and Health, Longyan University, Longyan, China, 4 School of Physical Education, Jining University, Jining, China

☯ These authors contributed equally to this work.
* 1003723595@qq.com

**Data Availability Statement:** All data relevant to the study are included in the article or uploaded as supplementary information. The data supporting

## Abstract

### Objective

This study conducts a rigorous meta-analysis of existing literature to rigorously examine the efficacy of Proprioceptive Neuromuscular Facilitation (PNF) in ameliorating functional deficits associated with Chronic Ankle Instability (CAI).

### Methods

Literature searches were conducted in multiple databases including China National Knowledge Infrastructure (CNKI), VIP, Wanfang, China Biology Medicine disc (CBM), PubMed, EBSCO (Medline, CINAHL, SPORTDiscus, and Rehabilitation & Sports Medicine Source), Embase, ScienceDirect, ProQuest, Cochrane Library, and Web of Science for randomized controlled trials assessing the effects of Proprioceptive Neuromuscular Facilitation interventions on patients with Chronic Ankle Instability. The publication timeframe spanned from the inception of each database until April 10, 2024. Meta-analysis was performed using STATA 12 software on the included studies.

### Results

① A total of 12 randomized controlled trials were included, encompassing 405 patients with Chronic Ankle Instability, demonstrating a generally high methodological quality of the literature. ② Meta-analysis results indicate that compared to the control group, Proprioceptive Neuromuscular Facilitation (PNF) significantly enhanced the balance ability of patients with Chronic Ankle Instability as measured by the Y Balance Test (YBT) (Weighted Mean Difference (WMD) = 3.61, 95% CI [2.65, 4.56], z = 7.42, P<0.001) and the Star Excursion Balance Test (SEBT) (WMD = 5.50, 95% CI [3.80, 7.19], z = 6.36, P<0.001), with improvement in all eight directions of SEBT balance ability surpassing that of the control group (P<0.05); muscle strength around the ankle (SMD) = 0.19, 95% CI [0.03, 0.36], z = 2.26, P = 0.024), with

the findings of this study are provided in the supplemental material.

**Funding:** The author(s) received no specific funding for this work.

**Competing interests:** The authors have declared that no competing interests exist.

both Plantar flexion and Dorsal flexion muscle strength improvements exceeding those of the control group (P<0.05); Visual Analog Scale (VAS) (WMD = -1.39, 95% CI [-1.72, -1.06], z = 8.23, P<0.001); Ankle instability questionnaire (WMD = 2.91, 95% CI [1.92, 3.89], z = 5.78, P<0.001).③Descriptive analysis results showed that the differences in Inversion Joint Position Sense and Dorsiflexion range of motion between the PNF and control groups were not statistically significant (P>0.05), however, the effects of PNF training persisted for a certain period even after cessation of treatment.

## Conclusion

Proprioceptive Neuromuscular Facilitation (PNF) can significantly improve balance, muscle strength, and pain in patients with Chronic Ankle Instability (CAI). While PNF has shown improvements in joint position sense and dorsiflexion range of motion for CAI patients, with effects that remain for a period thereafter, these improvements were not significantly different when compared to the control group. Further research is required to substantiate these specific effects.

## Introduction

Ankle sprain is a common musculoskeletal injury in sports [1], with up to 50% of patients with acute ankle sprain not seeking formal medical management. Repeated sprains can occur several months or years after the initial injury, developing into Chronic ankle instability (CAI) [2, 3]. CAI is characterized by a range of symptoms, which include reduced muscle strength in the area surrounding the ankle, delayed muscular activation, ligamentous laxity, compromised ankle stability, impaired proprioception, and restricted joint mobility [4]. Despite systematic rehabilitation attempts in the later stages, up to 70% of patients still exhibit various functional impairments within six months [5]. These deficiencies can significantly affect the physical activity levels and quality of life of individuals with CAI [6], potentially serving as principal risk factors for the early onset of post-traumatic osteoarthritis and joint degenerative diseases [7, 8]. Consequently, the rehabilitation of CAI patients has emerged as an important public health concern [9].

Current therapeutic approaches for CAI include both surgical and non-surgical interventions. The optimal surgical strategies and postoperative outcomes are subjects of ongoing debate [10, 11]. An increasing body of research and clinical practice underscores the manifest advantages of exercise-based non-surgical treatment in reducing the recurrence of ankle sprains and improving joint functionality in CAI patients [12]. Additionally, non-surgical treatments effectively circumvent the risks associated with surgery and reduce medical costs, thus garnering favor among patients and healthcare providers alike.

Proprioceptive Neuromuscular Facilitation (PNF) is a rehabilitative training methodology that employs proprioceptive stimuli, such as stretching, joint compression, traction, and resistive actions, combined with spiral and diagonal movement patterns, to foster the restoration of motor function and enhance neuromuscular responsiveness [13]. PNF has been shown to relax muscles, alleviate pain, increase joint range of motion, strengthen muscular force, and improve stability, bodily coordination, and postural control. These benefits play a significant and positive role in maintaining or enhancing physical ability and facilitating post-activity fatigue recovery [14, 15].

Principally utilized in the rehabilitation following neural damage [16, 17], PNF has demonstrated efficacy in ameliorating motor function impairments resulting from neurological injuries [18]. It has also been applied in musculoskeletal rehabilitation, especially vital in the recuperation from conditions such as low back pain, rotator cuff injuries, and frozen shoulder [19–21]. Recently, PNF has been adapted to the rehabilitation of individuals with CAI, though its effects on balance and functional recovery remain controversial [22–24]. To comprehensively assess the impact of PNF in the rehabilitation of CAI, this study employs a Meta-analysis to synthesize and scrutinize the effectiveness of PNF interventions on CAI rehabilitation outcomes. This objective evaluation aims to reinforce the role of PNF in CAI recovery and provide a more substantiated basis for clinical practice.

## Methods

### Protocol and registration

This meta-analysis has been registered in the PROSPERO system evaluation database (CRD42024533915) and completed according to the PRISMA list [25]. Since all data in this study were derived from experimental articles without direct recruitment or collection of patient information, ethical approval or consent statements were not necessary.

### Literature search

This meta-analysis retrieved databases such as China National Knowledge Infrastructure (CNKI), VIP, Wanfang, China Biomedical Literature Database (CBM), PubMed, EBSCO (Medline, CINAHL, SPORT Discus, Rehabilitation & Sports Medicine Source), Embase, Science Direct, ProQuest, Cochrane Library and Web of Science. The search time was set from the establishment of the database to April 2024, and the last search date was April 10, 2024. The search strategy involved a combination of subject words and free words, encompassing terms such as muscle stretching exercises, proprioceptive neuromuscular facilitation, PNF, PNF stretching, Proprioceptive Neuromuscular Facilitation Techniques, ankle sprain, ankle injury, ankle instability, chronic ankle instability, function ankle instability, CAI, FAI. Through the retrieval method of subject words combined with free words, manual retrieval is carried out and the references included in the literature are traced back to supplement and obtain more relevant documents. The retrieval strategy of the PubMed database is taken as an example (See Fig 1). (See S1 File. Tables for the complete search strategy of each database)

### Literature inclusion, exclusion criteria and outcome indicator

According to the PICOS principle, the inclusion, exclusion criteria and outcome indicator for the literature are as follows [26] (See Table 1):

### Literature screening and information extraction

Literature screening and data extraction were conducted independently by two researchers (Yin and Luo), who would cross-verify their findings. In case of discrepancies, discussions would ensue or a third researcher (Liu) would be consulted for resolution. During the literature screening process, titles were first reviewed to eliminate any obviously irrelevant studies, followed by a more detailed examination of abstracts and full texts to determine their suitability for inclusion. Should it be necessary, contact would be made with the authors of the original studies via email or telephone to acquire any ambiguous information. Data extraction primarily encompassed the following aspects:①Basic information of the included studies, such as study titles, lead authors, and the publication year;②basic characteristics of the study

```
#1 ankle sprain [MeSH Major Topic]
#2 chronic ankle instability [Text Word]
#3 CAI [Text Word]
#4 function ankle instability [Text Word]
#5 FAI [Text Word]
#6 ankle instability [Text Word]
#7 ankle injury [Text Word]
#8 #1 OR #2 OR #3 OR #4 OR #5 OR #6 OR #7
#9 muscle stretching exercises [MeSH Terms]
#10 proprioceptive neuromuscular facilitation [Text Word]
#11 Proprioceptive Neuromuscular Facilitation Techniques [Text Word]
#12 PNF [Text Word]
#13 reciprocal inhibition [Text Word]
#14 Spiral diagonal [Text Word]
#15 PNF stretching [Text Word]
#16 #9 OR #10 OR #11 OR #12 OR #13 OR #14 OR #15
#17 #8 AND #16
```

**Fig 1. Search strategy terms of PubMed.**

participants, including the number of cases and age across groups;③detailed descriptions of the interventions;④key elements for the assessment of risk of bias;⑤ outcome measures and result data of interest.

(The names of the data extractors and the date of data extraction are in S2 File)

## Assessment of risk of bias for study quality assessment

The quality assessment of the literature was independently conducted by two researchers (Lin and Luo), with cross-verification of results. In the event of a discrepancy in opinions, consultation with a third researcher (Liu) would be sought for resolution. Bias risk was independently

**Table 1. PICOS framework.**

| Parameter | Inclusion criteria | Exclusion criteria |
|---|---|---|
| P (population) | Adults (>18yrs) population from chronic ankle instability:<br>1) Subjects had at least one history of ankle sprain in the past 12 months, causing pain and swelling, and the time to lose normal function within 1 day or more<br>2) The affected ankle of subjects felt "soft leg", and/or repeatedly sprains and/or "un-stable" | Non-chronic ankle instability population, ankle surgery history, and animal research |
| I (intervention) | Proprioceptive neuromuscular facilitation(PNF), either alone or in conjunction with other measures | Non-clinical trials and studies without intervention designs |
| C (comparison) | Any alternative treatment includes standard care, placebo, or no treatment | |
| O (outcomes) | Functional scales, Balance tests, visual analog scale, Muscle strength, Joint Position Sense, Range of Motion | Studies from which data is not extracted or original data are inaccessible |
| S (Setting) | Language of Publication: Articles published in Chinese or English. | Non-Chinese/English documents; non-original research, such as reviews |

Note: Functional scales including the Ankle Joint Functional Assessment Tool Questionnaire (AJFAI) and the Cumberland Ankle Instability Tool (CAIT); Balance tests include the Star Excursion Balance Test (SEBT) in directions: Anterior (ANT), Anterolateral (ALAT), Anteromedial (AMED), Medial (MED), Posterior (POST), Lateral (LAT), Posterolateral (PLAT), and Posteromedial (PMED), as well as the Y Balance Test (YBT); Visual Analog Scale for Pain (VAS); Muscle strength assessments covering relative peak torque and maximum peak force during Plantar flexion, Dorsiflexion, Inversion, and Eversion; Inversion Joint Position Sense; and Ankle Dorsiflexion Range of Motion (DFROM).

evaluated using the second edition of the Cochrane Risk of Bias Tool (RoB2), which comprises five domains [27, 28]. The PEDro scale was utilized to assess the risk of bias and methodological quality of the included studies [29], with a scoring range of 0–10 points. Evaluation criteria dictated that one point be awarded for each of the items 2–11, with a total possible score of 10 points. Studies scoring $\geq 6$ points are considered high quality, those with 4–5 points are deemed moderate quality, and scores of $\leq 3$ are rated as low quality [30].

## Statistical analysis

Statistical analyses were conducted using STATA.12 (Stata Corporation, College Station, TX, USA) to process the data metrics from the included studies. It can quickly and efficiently identify publication bias and accurately and reliably generate meta-analysis plots [31]. For continuous outcomes across all included studies, analysis was conducted using mean difference (MD) and 95% confidence interval (CI). When the units of measurement were identical, weighted mean difference (WMD) was selected as the measure of effect, while if the units of measurement varied, standard mean difference (SMD) was utilized as the measure of effect. Analysis was completed with a 95% confidence interval (CI). Heterogeneity in the data was assessed using chi-squared and $I^2$ tests. An $I^2$ of 0 indicates no inconsistency among the included studies, while a higher $I^2$ value implies greater heterogeneity. A significance threshold for heterogeneity was set at $P \leq 0.1$ and $I^2 > 50\%$. For $P > 0.1$ and $I^2 < 50\%$, a fixed-effects model was employed for the meta-analysis, whereas other scenarios utilized a random-effects model [32]. Studies with $\geq 5$ entries underwent Egger's test for publication bias and sensitivity analyses, with $P < 0.05$ denoting statistical significance [33].

## Results

### Study selection and characteristics

Following the database search, an initial retrieval yielded 288 literatures, comprising 28 in Chinese and 260 in English, with no additional literature obtained through other channels. Subsequently, EndNote X9 was utilized to merge and remove duplicates, resulting in the exclusion of 152 redundant literatures. Titles and abstracts were reviewed, leading to the selection of 136 literatures. Full-text readings further narrowed the field to 32 literatures. According to the inclusion and exclusion criteria, ultimately 12 literatures were determined to be suitable for inclusion in the analysis [14, 22–24, 34–41] (The literature excluded after reading the full text is detailed in S3 File).

 The literature screening process and results are shown in Fig 2. According to the inclusion and exclusion criteria, a total of 12 literatures [14, 22–24, 34–41] were selected, encompassing 405 patients with CAI. Among these, six literatures [14, 22, 35–38] utilized PNF as the primary intervention in the experimental group. Four literatures reported the experimental group receiving PNF combined with core stability and conventional rehabilitation training [34, 39–41]. The control interventions included conventional rehabilitation training in four literatures [24, 34, 39, 40], no intervention in two literatures [14, 38], and bicycle training in two literatures [35, 36]. All included literatures were published in the last decade, accounting for 100% of the selection (2015 to 2023). Basic information about the included literature can be found in Table 2, and detailed characteristics of the included literature are presented in Table 3.

### Quality and risk of bias

For the 12 included literatures, bias risk was assessed using the second edition of the Cochrane Risk of Bias Tool (RoB2), with all articles deemed to be at low risk. According to the PEDro

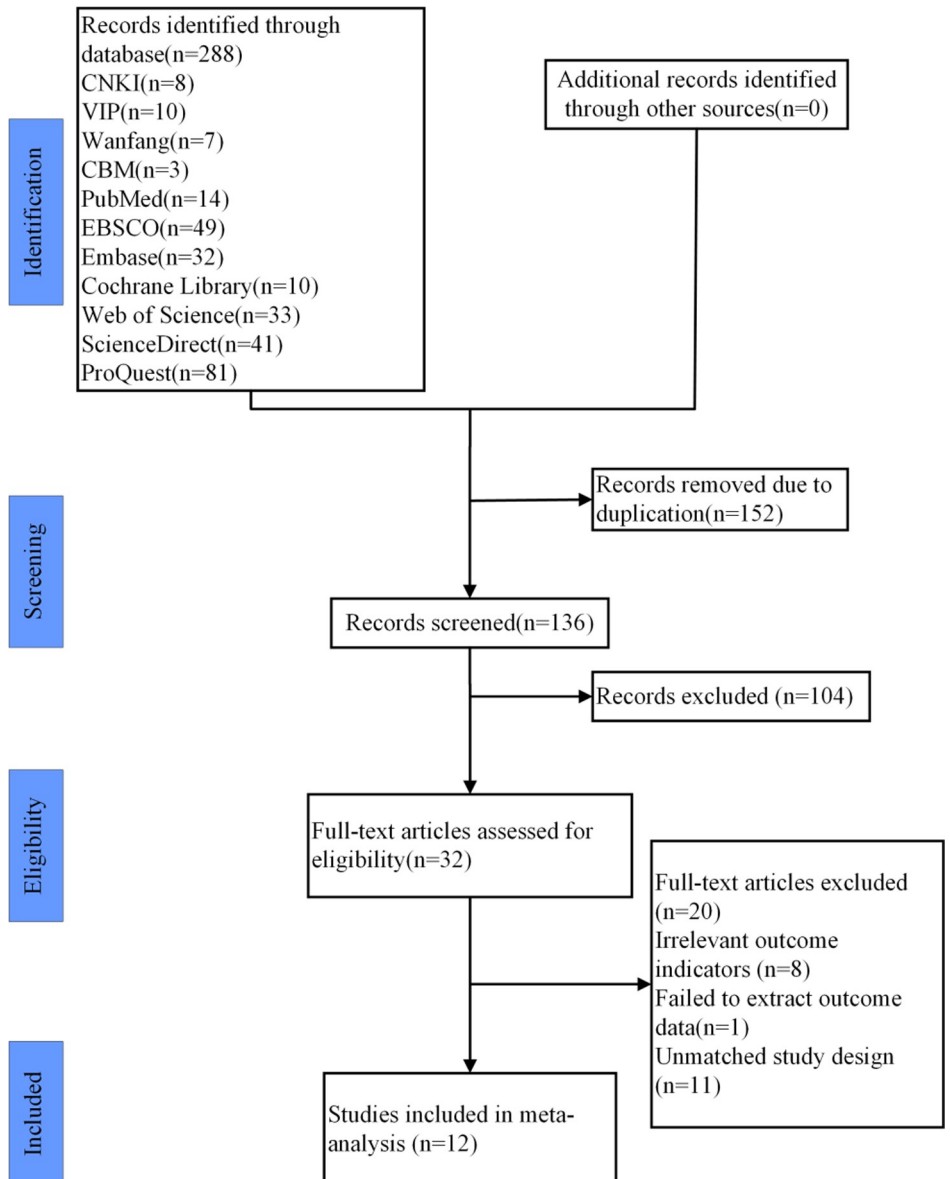

**Fig 2. Study selection represented by PRISMA flowchart.**

scoring system, all included articles were of medium to high quality (ranging from 5 to 10), with an average score of 6.9, indicating that the methodological quality of the studies included in the reports was high-quality. This suggests that the overall quality of the literature is high (See Fig 3/Table 4).

The Cochrane Risk of Bias Tool (RoB2) is detailed in S4 File.

## Effect of PNF on balance

YBT: Three literatures involving 132 patients were included, reporting on the effect of PNF on CAI patients' YBT scores. The data from these studies showed no heterogeneity: p = 0.412, $I^2$ = 0%, hence a random-effects model was applied for analysis. The meta-analysis results indicated that PNF intervention led to a significant improvement in YBT scores compared to the control

**Table 2. Basic information about the included literature.**

| Research | Country | Research type | Subject type | Age(T/C, year) | Sample size (T/C,n) | Gender (Male/female,n) | Research Group (T/C) |
|---|---|---|---|---|---|---|---|
| Linhua X 2021 | China | RCT | CAI | 21.31±3.96/ 23.46±4.65 | 25/25 | 24/26 | PNF+①+②/② |
| Mengfan S2017 | China | RCT | CAI | 21.5±1.27/21.6±1.35 | 10/10 | 8/12 | PNF+①+②/② |
| Yuduo L2022 | China | RCT | CAI | 20.05±2.27/19.71±0.67 | 8/9 | 17/0 | PNF/③ |
| Renyi 2023 | China | RCT | CAI | 37.98±7.34/32.56±5.13 | 24/24 | 25/23 | PNF+②+④/② |
| Xuepeng X2020 | China | RCT | CAI | 23.33±3.19/22.28±3.16 | 19/19 | 33/25 | PNF+①+②/② |
| Jiesong Y 2023 | China | RCT | CAI | 20.3±1.8/20.4±1.7 | 17/17 | 19/15 | PNF+⑤/②+⑤ |
| Alahmari K A 2020 | Saudi Arabia | RCT | CAI | 25.7±5.6/25.9±6.2 | 20/20 | - | PNF/No intervention |
| Lazarou L 2017 | Greece | RCT | CAI | 22±3.8/22±1.8 | 10/10 | 6/14 | PNF/⑤ |
| Hall E A2015 | America | RCT | CAI | 18.9±1.3/20.5±2.1 | 13/13 | 12/14 | PNF/No intervention |
| Hall E A2018 | America | RCT | CAI | 24.6±7.7/24.8±9.0 | 13/13 | 14/12 | PNF /⑥ |
| Hall E A2018 | America | RCT | CAI | 24.6±7.7/24.8±9.0 | 13/13 | 14/12 | PNF /⑥ |
| PANG Min2019 | China | RCT | CAI | 21.34±0.83/21.54±0.87 | 30/30 | 31/29 | PNF+①+②/② |

Note: RCT(Randomized controlled trial); CAI(Chronic ankle instability);①core stability training;②routine rehabilitation training;③Whole Body Vibration Training;④Neuromuscular Electrical Stimulation;⑤Balance training;⑥bicycle

group (WMD = 3.61, 95%CI(2.65, 4.56), z = 7.42, P<0.001), suggesting statistically significant outcomes(See Fig 4).

SEBT: Four literatures, encompassing 144 patients, reported on the effect of PNF on CAI patients' SEBT scores. The included studies showed significant heterogeneity: p = 0.00, $I^2$ = 94.4%, therefore, a random-effects model was used for analysis. The meta-analysis results demonstrated that PNF intervention improved SEBT more effectively than the control group (WMD = 5.50, 95%CI (3.80, 7.19), z = 6.36, P<0.001), indicating statistically significant results (See Fig 5). Heterogeneity was substantially attributed to the study by Jiesong Y [23]; upon its exclusion, the remaining three articles showed no significant statistical heterogeneity: p = 0.092, $I^2$ = 29.0%, and a fixed-effects model was applied; the meta-analysis indicated (WMD = 2.47, 95%CI(1.85, 3.09), z = 7.81, P<0.001) (The meta-graph on exclusion heterogeneity research is in S5 File). Further subgroup analysis of the eight directions of SEBT (Anterior (ANT), Anterolateral (ALAT), Anteromedial (AMED), Medial (MED), Posterior (POST), Lateral (LAT), Posterolateral (PLAT), and Posteromedial (PMED)) showed:①ANT (WMD = 1.96, 95%CI(0.42,3.49), P = 0.012);②ALAT (WMD = 1.90, 95%CI(0.47, 3.32), P = 0.009);③AMED (WMD = 3.44, 95%CI(1.21,5.67), P = 0.003);④MED (WMD = 2.93, 95% CI(1.18,4.68), P = 0.001);⑤POST (WMD = 3.12, 95%CI(0.88,5.36), P = 0.006)⑥LAT (WMD = 2.82, 95%CI(0.81,4.82), P = 0.006);⑦PLAT (WMD = 2.57, 95%CI(0.88,4.26), P = 0.003); ⑧PMED (WMD = 2.20, 95%CI(0.54, 3.87), P = 0.009). These assessments demonstrated that PNF intervention improved balance in all eight directions of the SEBT in subjects with Chronic Ankle Instability compared to the control group.

## Effect of PNF on muscle strength

Muscle strength was assessed using both an isokinetic dynamometer and an isometric hand-held dynamometer. The former was used to measure the relative peak torque of the muscles [42], whereas the latter evaluated the maximum peak force [14]. Higher values of relative peak torque and maximum peak force indicate greater muscle strength. Since the units of measurement for the two methods differ, the standardized mean difference (SMD) was chosen for data synthesis. Four literatures reported the effects of PNF on Muscle Strength, with no significant heterogeneity observed among the included studies: p = 0.075, $I^2$ = 36.0%. Therefore, a fixed-

**Table 3. Detailed characteristics of the included literature.**

| Research | Inclusion criteria | PNF intervention | Outcome | Adverse reaction | conclusion |
|---|---|---|---|---|---|
| Linhua X 2021 | 1. Patients with lateral ankle ligament sprains who have not received rehabilitation training; 2. Have not undergone ankle fracture surgery history. | Diagonal 2-fexion (D2-F) pattern of lower limb flexion and diagonal 2-extension (D2-E) pattern of lower limb extension, 15 times/group, 3 groups /d, Once every two days, 1 month. | I/II/III/IV | Not report | PNF significantly improves balance in patients with CAI, but the improvement in muscle strength and positional sense is not superior to the control group. |
| Mengfan S2017 | 1. History of unilateral ankle sprain, with a feeling of ankle instability; 2. Negative anterior drawer test/talar tilt test results; 3. Have not received rehabilitation training; 4. No history of ankle fracture surgery. | Diagonal 2-fexion (D2-F) pattern of lower limb flexion and diagonal 2-extension (D2-E) pattern of lower limb extension, 15 times/group, 3 groups /d, 23 days, 12 times. | I/II/III/IV | Not report | PNF significantly improves balance, muscle strength, and positional sense in patients with CAI. |
| Yuduo L2022 | 1. CAIT score ≤24 points; 2. History of unilateral ankle sprain; 3. Negative anterior drawer test/talar tilt test results; 4. No history of ankle fracture surgery. | Diagonal 2-fexion (D2-F) pattern of lower limb flexion and diagonal 2-extension (D2-E) pattern of lower limb extension, 20min/ time, 3 times/week, 6 weeks. | I | Total 5 subjects were excluded. | There was no significant difference in balance ability between the two groups. |
| Renyi 2023 | 1. History of unilateral ankle sprain; 2. No history of ankle fracture surgery. | combination of isotonic, dynamic reversals and stabilizing reversals, Each action is done in sets of 10,3 groups/time, 3 times/week, 8 weeks. | I/II/IV/V | Not report | PNF significantly improves balance, muscle strength, and pain in patients with CAI. |
| Xuepeng X2020 | - | Diagonal 2-fexion (D2-F) pattern of lower limb flexion and diagonal 2-extension (D2-E) pattern of lower limb extension, 15 times/group, 3 groups /d, 12 times. | IV | Not report | PNF significantly improves self-assessment in patients with CAI. |
| Jiesong Y 2023 | 1. History of unilateral ankle sprain, with a feeling of ankle instability; 2. AJFAT score <26 points; 3. Negative anterior drawer test/talar tilt test results; 4. No history of ankle fracture surgery. | Diagonal 2-fexion (D2-F) pattern of lower limb flexion and diagonal 2-extension (D2-E) pattern of lower limb extension, 15 times/group,2 groups /d, 6 weeks, 21 times. | I/IV | Not report | PNF significantly improves balance in patients with CAI. |
| Alahmari K A 2020 | 1. History of unilateral ankle sprain; 2. No history of ankle fracture surgery. | hold-relax technique, 30min/ time, 4 times/week, 3 weeks. | I/II/III/ IV/V/VI | Not report | PNF significantly improves balance, muscle strength, positional sense, range of motion, and pain in patients with CAI. The treatment effect was sustained. |
| Lazarou L 2017 | 1. History of unilateral ankle sprain; 2. No history of ankle fracture surgery. | rhythmic stabilization (RS) and combination of isotonics (COI), 6 weeks, 10 times. | III/IV/VI | Not report | PNF significantly improves positional sense, dorsiflexion range of motion, and pain in patients with CAI. The treatment effect was sustained, but there was no significant difference compared with the control group. |
| Hall E A2015 | 1.IdFAI score≥11 points, 2. History of unilateral ankle sprain; 3. No history of ankle fracture surgery. | Diagonal 2-fexion (D2-F) pattern of lower limb flexion and diagonal 2-extension (D2-E) pattern of lower limb extension, 3 times/week, 6 weeks. | I/II/V | Not report | PNF significantly improves balance, muscle strength and pain in patients with CAI. |
| Hall E A2018 | 1.IdFAI score≥11 points, 2. History of unilateral ankle sprain; 3. No history of ankle fracture surgery. | Spiral and diagonal pattern,3 times/week, 6 weeks. | I/II | Not report | PNF significantly improves balance and muscle strength in patients with CAI. |
| Hall E A2018 | 1.IdFAI score≥11 points, 2. History of unilateral ankle sprain; 3. No history of ankle fracture surgery. | Spiral and diagonal pattern,3 times/week, 6 weeks. | IV/V | Not report | PNF significantly improves self-assessment and quality of life in patients with CAI. |
| PANG Min2019 | 1. History of unilateral ankle sprain; 2. No history of ankle fracture surgery. | Combination of isotonics (COI) and Reversals rhythmic stabilization,10 weeks. | IV | Not report | PNF significantly improves self-assessment and ankle function in patients with CAI. |

Note: I:Balance test (YBT/SEBT);II:Muscle strength;III:Ankle position perception (JPS);IV:Ankle Questionnaire (AJFAT/CAIT/FADI);V:Visual Analogue Pain scale (VAS);VI:Dorsiflexion range of motion. "-" is not mentioned

**Fig 3. Risk of bias assessed using RoB 2 tool.**

effect model was applied for analysis. The results indicated that Muscle Strength improvements post-PNF intervention were superior to the control group (SMD = 0.19, 95%CI(0.03, 0.36), z = 2.26, p = 0.024), which is statistically significant. Subgroup analysis for muscle strength in the four directions of ankle movement showed the following: for Plantar flexion (SMD = 0.39, 95%CI(0.09,0.68), P = 0.010), Dorsal flexion (SMD = 0.33, 95%CI(0.04, 0.63), P = 0.026), Varus (SMD = 0.00, 95%CI(-0.40,0.40), P = 1.00), and Eversion (SMD = -0.24, 95% CI(-0.64,0.17), P = 0.249). The results demonstrate that PNF significantly improved Muscle Strength in the Plantar and Dorsal flexion for CAI patients compared to the control group, while no superior improvement was observed in the Varus and Eversion directions (see Fig 6).

**Table 4. PEDro scale score of included literature.**

| Study | PEDro scale | | | | | | | | | | | |
|---|---|---|---|---|---|---|---|---|---|---|---|---|
| | ① | ② | ③ | ④ | ⑤ | ⑥ | ⑦ | ⑧ | ⑨ | ⑩ | ⑪ | score |
| Linhua X 2021 | Yes | 1 | 1 | 1 | | | | 1 | 1 | 1 | 1 | 7 |
| Mengfan S2017 | Yes | 1 | 1 | 1 | | | | 1 | 1 | 1 | 1 | 7 |
| Yuduo L2022 | Yes | 1 | 1 | 1 | | | | 1 | 1 | 1 | 1 | 7 |
| Renyi 2023 | Yes | 1 | 1 | 1 | | | | 1 | 1 | 1 | 1 | 7 |
| Xuepeng X2020 | Yes | 1 | 1 | 1 | | | | 1 | 1 | 1 | 1 | 7 |
| Jiesong Y 2023 | Yes | 1 | 1 | 1 | | | | 1 | 1 | 1 | 1 | 7 |
| Alahmari K A 2020 | Yes | 1 | 1 | 1 | 1 | | 1 | 1 | 1 | 1 | 1 | 9 |
| Lazarou L 2017 | Yes | 1 | 1 | 1 | 1 | | 1 | 1 | 1 | 1 | 1 | 9 |
| Hall E A2015 | Yes | 1 | | 1 | | | | 1 | 1 | 1 | 1 | 6 |
| Hall E A2018 | Yes | 1 | | 1 | | | | 1 | 1 | 1 | 1 | 6 |
| Hall E A2018 | Yes | 1 | | 1 | | | | 1 | 1 | 1 | 1 | 6 |
| PANG Min2019 | Yes | | | 1 | | | | 1 | 1 | 1 | 1 | 5 |

Note:①Whether the inclusion conditions of subjects are clear;②Randomly grouping;③Blind grouping;④Baseline consistency of main prognostic indicators;⑤The subjects were blinded;⑥**Training** is blind;⑦All the evaluators of at least one main result are blind;⑧Measure at least one main result for more than 85% of the subjects;⑨Assign subjects according to the distribution scheme to receive treatment or control conditions;⑩Report at least one main result of inter-group statistical results;⑪Provide at least one point measurement and variation measurement of the main result.

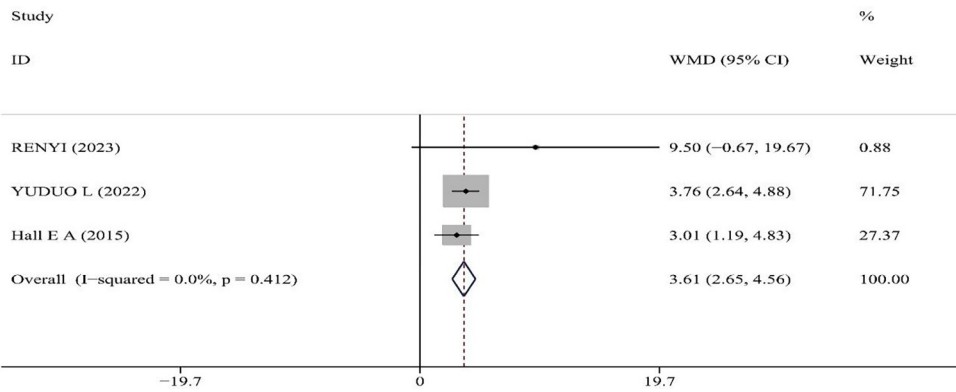

**Fig 4. Meta-analysis of the effect of PNF on Y balance test.**

## Effect of PNF on self-assessment

VAS: Four literatures reported on the effects of PNF on VAS, with no heterogeneity observed among the included studies (p = 0.106, I$^2$ = 51%). Therefore, a fixed-effects model was used for

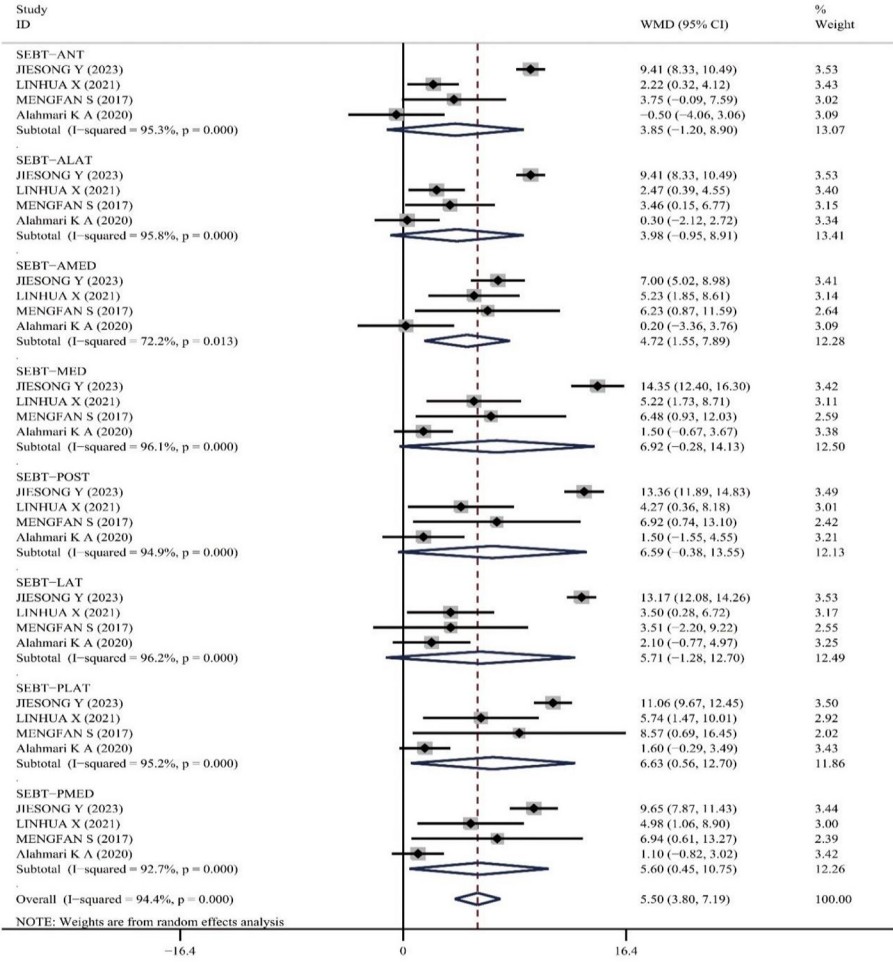

**Fig 5. Meta-analysis of the effect of PNF on star excursion balance test.**

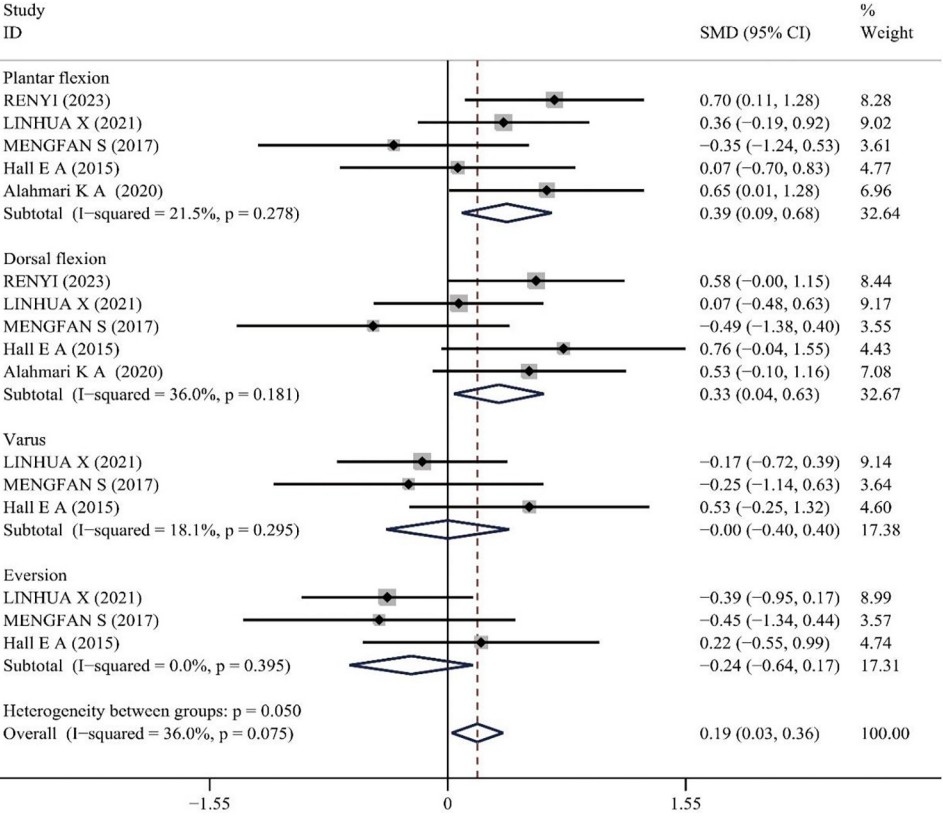

**Fig 6. Meta-analysis of the effect of PNF on muscle strength.**

the analysis. The results demonstrated that the improvement in VAS following PNF intervention was superior to that of the control group (WMD = -1.39, 95% CI [-1.72, -1.06], z = 8.23, p<0.001), and the results were statistically significant (see Fig 7).

Ankle Instability Questionnaire: Seven literatures involving 310 patients were included to assess the effect of PNF on the Ankle Instability Questionnaire in patients with chronic ankle

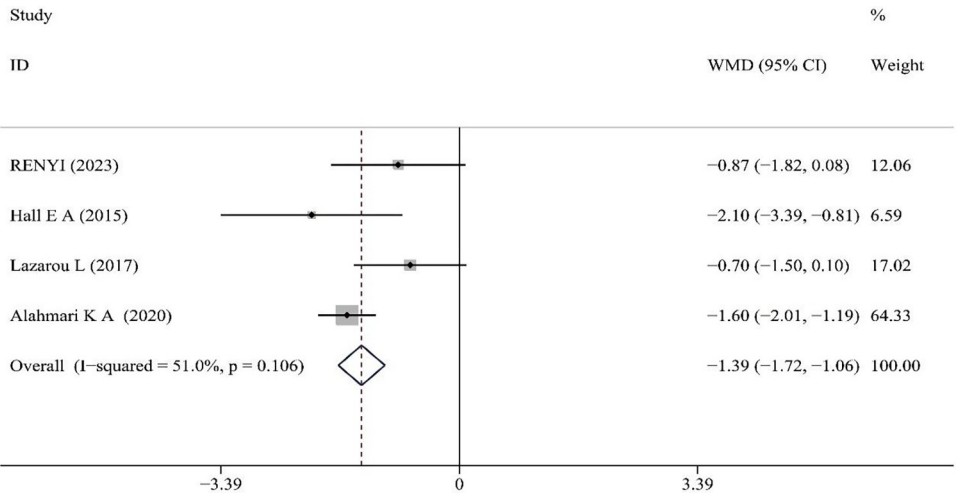

**Fig 7. Meta-analysis of the effect of PNF on VAS.**

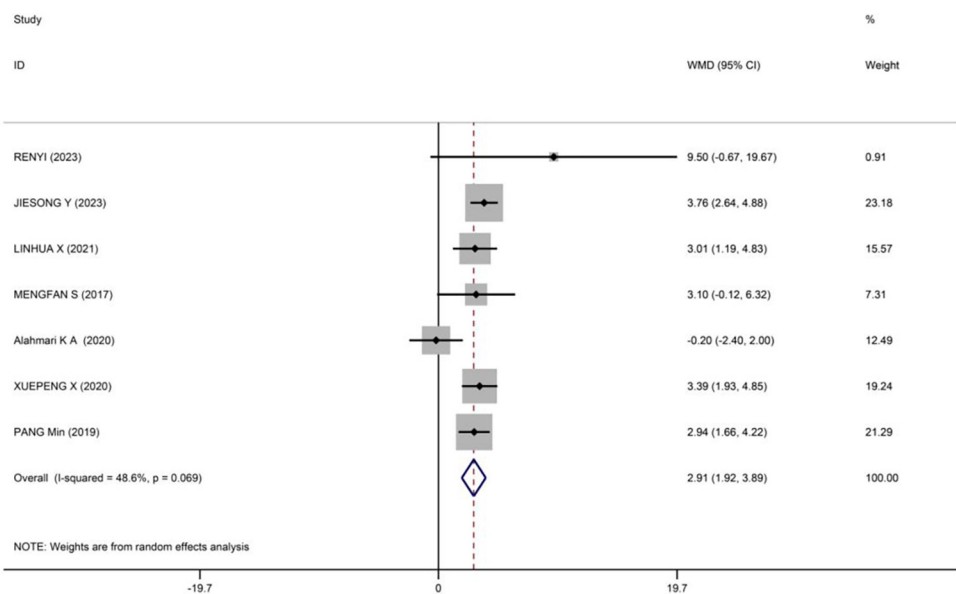

**Fig 8. Meta-analysis of the effect of PNF on Ankle instability questionnaire.**

instability (CAI). There was heterogeneity among the included studies (p = 0.069, $I^2$ = 48.6%), so a random-effects model was used for the analysis. The meta-analysis results indicated that PNF intervention significantly improved the Ankle Instability Questionnaire scores compared to the control group (WMD = 2.91, 95% CI [1 92, 3.89], z = 5.78, p<0.001) (see Fig 8). The study by Alahmari KA [38] contributed significantly to the heterogeneity. Upon exclusion of this literature, the remaining six literatures showed no statistical heterogeneity (p = 0.781, $I^2$ = 0%), and a fixed-effects model was then used for the analysis. The meta-analysis results demonstrated a significant improvement in the Ankle Instability Questionnaire scores with PNF intervention (WMD = 3.36, 95% CI [2.70, 4.02], z = 9.95, p<0.001) (The meta-graph on exclusion heterogeneity research is in S5 File).

### Descriptive analysis results

**Effect of PNF on inversion joint position sense.**   Only two literatures reported on the effects of PNF on Inversion Joint Position Sense, which did not meet the criteria for a meta-analysis, thus descriptive analysis was used. Research by Linhua X and Mengfan S, et al. [34, 40] found that after a period of PNF training in patients with CAI, the difference in Inversion Joint Position Sense between the experimental group and the control group was not statistically significant (P>0.05).

**Effect of PNF on Dorsiflexion range of motion.**   Only two literatures reported on the effects of PNF on the Dorsiflexion range of motion, which did not meet the criteria for a meta-analysis, thus a descriptive analysis was employed. Research conducted by Lazarou L, et al. [37] found that after 4 weeks of PNF training for CAI patients, and a study by Alahmari K A, et al. [38] which provided training for CAI patients four times a week over a continuous three-week period, improvements in Dorsiflexion range of motion were not superior to the control group. However, follow-up results from both studies demonstrated that the effectiveness of PNF training was maintained for some time even after the cessation of treatment.

**Publication bias.**   Egger's test was utilized to analyze publication bias for the following indicators: the Ankle Instability Questionnaire, Plantar Flexion Muscle Strength, and Dorsal

Flexion Muscle Strength. Results revealed no significant publication bias for these indicators, with the Ankle Instability Questionnaire (t = -0.15, P = 0.889, P>0.05), Plantar Flexion (t = -2.56, P = 0.083, P>0.05), and Dorsal Flexion (t = -0.47, P = 0.669, P>0.05) (Publication bias is in the S6 File).

## Discussion

Chronic Ankle Instability (CAI) is defined as the structural or functional alterations in the ankle and surrounding joint tissues due to recurrent sprains [43], resulting in sensations of instability, movement restrictions, intermittent pain, muscle weakness around the ankle, ligament laxity, and other functional impairments [44], severely impacting the physical health and quality of life of active individuals. Thus, the main objectives of the intervention are to enhance the muscle strength surrounding the ankle, improve proprioception, and better balance capabilities. From an evidence-based medicine perspective, this study incorporates 12 literatures, covering a total of 405 CAI patients. The meta-analysis results reveal that proprioceptive neuromuscular facilitation techniques significantly improve balance, muscle strength, and joint position sense, and alleviate pain in patients with CAI.

Chronic Ankle Instability (CAI) often manifests as abnormalities in postural control to varying degrees [45]. After an initial ankle sprain, the anterior talofibular ligament, posterior talofibular ligament, and calcaneofibular ligament may become lax to varying extents, or even partially torn, leading to damage of proprioceptors that perceive joint speed and position during movement [46]. This results in abnormal transmission of postural control pathway information, subsequently causing postural control disorders and a decline in balance [47]. The Y Balance Test (YBT, ICC = 0.88–0.99) [48] and the Star Excursion Balance Test (SEBT, ICC = 0.85–0.96) [49] are dynamic tests requiring strength, flexibility, and proprioception. They are commonly used to assess dynamic body balance capabilities and lower limb dynamic postural control in patients with CAI [50]. Meta-analysis results indicate that PNF shows a significant advantage over conventional training in improving postural control abilities in patients with CAI (P<0.05). The results suggest that PNF techniques can improve lower limb postural control, aligning with previous findings [23, 38]. PNF techniques often involve dynamic stretching, joint compression, traction, and isometric contractions, which stimulate joint proprioceptors, increase neuronal excitability and conduction velocity, and improve the efficiency of neuromuscular conduction pathways, thereby enhancing motor control [51]. Furthermore, PNF techniques, through repetitive and complex multi-planar movements, involvement and coordination of multiple joints, muscle groups, and sensory inputs (visual, tactile, and proprioceptive feedback), stimulate the sensory processing centers in the brain and spinal cord. This improves the synergy and synchrony between muscles, strengthens the feedback regulation loop of the nervous system, and enhances the central nervous system's perception and response capabilities to movement [52, 53].

Muscle strength is foundational for maintaining postural control and balance stability. During high-intensity activities such as walking, running, and jumping, robust muscle strength can effectively buffer the ground reaction forces [54]. Patients with CAI commonly exhibit a decrease in muscle strength during inversion and eversion of the ankle, as well as prolonged muscle response times [55]. If this persists, it can lead to muscle strength asymmetry. Asymmetry often results in excessive muscle compensation and increases the risk of recurrent sports injuries [56]. Therefore, strengthening the muscle strength around the ankle is key to preventing the recurrence of ankle sprains. Meta-analysis results demonstrate that PNF has a significant advantage over conventional training in improving the muscle strength around the ankle in CAI patients, with particularly noticeable improvements in Plantar flexion and Dorsal

flexion (P<0.05). Specific movement patterns in PNF techniques, such as the D2 pattern, and the use of Hold-Relax, rhythmic stabilization (RS), and combination of isotonics (COI), can thoroughly activate and strengthen the musculature surrounding the ankle joint [57]. The repetitive static and dynamic exercises along spiral diagonals in PNF techniques not only stimulate the muscles and intra-articular receptors but also activate the motor-evoked potentials in the prefrontal cortex of the brain [58]. This increases the electromyographic signal output, encouraging the recruitment of more neuromuscular units, enhancing the rate and synchronicity of muscle fiber recruitment, and activating the coordination of the ankle's surrounding musculature (such as the peroneal muscle group, tibialis anterior, and triceps surae). This helps maintain joint stability during prolonged activities [37]. Compared to other interventions, PNF training also involves additional sensory stimuli (tactile, visual, or verbal) to better facilitate neuromuscular responses and enhance the subject's neural control of the muscles [59].

CAI patients have a high recurrence rate and are often overlooked in everyday settings. Regularly conducting subjective assessments and descriptions of CAI patients using ankle function scales to judge pain, loss of control, and motor function status can help understand the effectiveness of CAI rehabilitation and the implementation and improvement of rehabilitation strategies [60]. Currently, there is still a lack of a "gold standard" for evaluating CAI scales. Researchers make subjective judgments about CAI using various scales developed to evaluate ankle function, and there is no unified standard for the reliability and validity of these scales [61]. The functional scales used in the included studies of this article, namely AJFAT (ICC = 0.94), FADI (ICC = 0.84), and CAIT (ICC = 0.95), all show good internal reliability. The meta-analysis results revealed that ankle joint function significantly improved following CAI intervention through PNF training (P<0.05); PNF can effectively alleviate the sensation of pain around the ankle in CAI patients(p<0.001). This may be attributed to the fact that PNF training, through dynamic movements and isometric contractions, promotes blood circulation around the ankle joint, accelerating the repair and regeneration of damaged tissues, thereby reducing inflammation and associated pain [62]. PNF can alleviate pain, help patients avoid discomfort during movement and daily activities, improve joint function and quality of life, and reduce the risk of secondary injuries.

Joint Position Sense refers to the ability to perceive joint positions, which is essential for executing precise movements, maintaining correct posture and balance, and preventing excessive joint extension or injury [63]. In CAI patients, repeated ankle sprains can lead to the impairment of internal sensors (such as muscle spindles, Golgi tendon organs, and joint receptors), resulting in abnormal ankle proprioception [64]. The results of this study indicate that PNF can improve the Joint Position Sense in CAI patients, but its effectiveness is not superior to conventional intervention measures. A meta-analysis conducted by AKASAKI H et al. [65] on the effects of PNF on joint position sense showed that PNF did not provide significant advantages, which is consistent with the results of this study. CAI patients often experience anterior displacement of the talus after a sprain, which can prevent effective posterior sliding during movement, leading to an inability to achieve a stable closed position of the joint and limiting dorsiflexion activity [66]. The study results show that PNF can improve dorsiflexion mobility in CAI patients, though not superior to the control group, it is noteworthy that the improvement effects can be maintained for a certain period after PNF training.

## Study limitations

The limitations of this meta-analysis include: ① Only 12 literatures were included, with a relatively small data sample size, and bias analysis was not conducted, which may affect the

conclusions; ② Only two of the included studies conducted follow-ups after the training, the remaining nine did not follow up on the intervention effects;③ The experimental designs of the included studies, assessment indicators, and subject characteristics varied, which impacted the reference value of the results; ④ The control groups lacked a unified standard, affecting the accuracy of the results.

## Conclusions

In conclusion, the current meta-analysis demonstrates that proprioceptive neuromuscular facilitation (PNF) significantly enhances balance, and muscle strength, and alleviates pain in patients with Chronic Ankle Instability (CAI). PNF has been observed to improve joint position sense and dorsiflexion range of motion among CAI patients, with sustained effects observed for a period following the intervention. In clinical treatment, PNF training is recommended for improving pain, balance ability, and muscle strength in patients with CAI. However, additional therapeutic approaches should be incorporated if patients with CAI experience significant mobility limitations.

For future research directions, we propose the following: (1) To optimize research protocols and experimental designs by using PNF as the sole variable in the experimental group, and including other interventions, either without PNF or with sham PNF, in the control group to determine the combined effects of PNF and other interventions; (2) To increase the sample size and conduct medium- and long-term follow-ups on CAI patients treated with PNF to assess whether the improvements are sustained over time; (3) To further conduct high-quality randomized clinical trials in clinical settings to provide more evidence-based support for clinical application.

## Supporting information

**S1 File. Search strategy for each database.**
(DOCX)

**S2 File. Data extraction table.**
(DOCX)

**S3 File. Excluded literature.**
(DOCX)

**S4 File. Risk of Bias Tool (RoB2).**
(XLSX)

**S5 File. Exclusion heterogeneity research.**
(DOCX)

**S6 File. Publication bias.**
(DOCX)

## Acknowledgments

We express our gratitude to all professors and colleagues for their unwavering support throughout this project. We also extend our sincere appreciation to the reviewers for their rigorous evaluation of our work and their invaluable recommendations for further improvement.

## Author Contributions

**Conceptualization:** Yikun Yin, Jialin Wang.

**Data curation:** Yikun Yin, Yinghang Luo, Yongsheng Liu.

**Formal analysis:** Yikun Yin, Qihan Lin, Yinghang Luo.

**Validation:** Yikun Yin, Yinghang Luo.

**Writing – original draft:** Yikun Yin, Jialin Wang.

**Writing – review & editing:** Yikun Yin, Jialin Wang, Junzhi Sun.

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
