## [Decision Letter · Decision Letter 0]

30 Aug 2024

PONE-D-24-25236Effect of proprioceptive neuromuscular facilitation on patients with chronic ankle instability: A systematic review and meta-analysisPLOS ONE

Dear Dr. Yin,

Thank you for submitting your manuscript to PLOS ONE. After careful consideration, we feel that it has merit but does not fully meet PLOS ONE’s publication criteria as it currently stands. Therefore, we invite you to submit a revised version of the manuscript that addresses the points raised during the review process.

Please submit your revised manuscript by Oct 14 2024 11:59PM. If you will need more time than this to complete your revisions, please reply to this message or contact the journal office at plosone@plos.org. Please include the following items when submitting your revised manuscript:A rebuttal letter that responds to each point raised by the academic editor and reviewer(s). You should upload this letter as a separate file labeled 'Response to Reviewers'.A marked-up copy of your manuscript that highlights changes made to the original version. You should upload this as a separate file labeled 'Revised Manuscript with Track Changes'.An unmarked version of your revised paper without tracked changes. You should upload this as a separate file labeled 'Manuscript'.If applicable, we recommend that you deposit your laboratory protocols in protocols.io to enhance the reproducibility of your results. Protocols.io assigns your protocol its own identifier (DOI) so that it can be cited independently in the future. For instructions see: https://journals.plos.org/plosone/s/submission-guidelines#loc-laboratory-protocols. Additionally, PLOS ONE offers an option for publishing peer-reviewed Lab Protocol articles, which describe protocols hosted on protocols.io. Read more information on sharing protocols at https://plos.org/protocols?utm_medium=editorial-email&utm_source=authorletters&utm_campaign=protocols.

We look forward to receiving your revised manuscript.

Kind regards,

Renato S. Melo, PhD

Academic Editor

PLOS ONE

Journal Requirements:

1. When submitting your revision, we need you to address these additional requirements. Please ensure that your manuscript meets PLOS ONE's style requirements, including those for file naming. The PLOS ONE style templates can be found at https://journals.plos.org/plosone/s/file?id=wjVg/PLOSOne_formatting_sample_main_body.pdf and https://journals.plos.org/plosone/s/file?id=ba62/PLOSOne_formatting_sample_title_authors_affiliations.pdf 2. We note that the grant information you provided in the ‘Funding Information’ and ‘Financial Disclosure’ sections do not match.  When you resubmit, please ensure that you provide the correct grant numbers for the awards you received for your study in the ‘Funding Information’ section. 3. Please update your submission to use the PLOS LaTeX template. The template and more information on our requirements for LaTeX submissions can be found at http://journals.plos.org/plosone/s/latex. 4. In the online submission form, you indicated that your data is available only on request from a third party. Please note that your Data Availability Statement is currently missing [the name of the third party contact or institution / contact details for the third party, such as an email address or a link to where data requests can be made]. Please update your statement with the missing information.  5. Please include captions for your Supporting Information files at the end of your manuscript, and update any in-text citations to match accordingly. Please see our Supporting Information guidelines for more information: http://journals.plos.org/plosone/s/supporting-information.

Reviewers' comments:

Reviewer's Responses to Questions

**Comments to the Author**

1. Is the manuscript technically sound, and do the data support the conclusions?

Reviewer #1: Yes

Reviewer #2: Yes

2. Has the statistical analysis been performed appropriately and rigorously? 

Reviewer #1: Yes

Reviewer #2: Yes

3. Have the authors made all data underlying the findings in their manuscript fully available?

Reviewer #1: Yes

Reviewer #2: Yes

4. Is the manuscript presented in an intelligible fashion and written in standard English?

Reviewer #1: Yes

Reviewer #2: Yes

5. Review Comments to the Author

Reviewer #1: General Comments:

The manuscript, "Effect of proprioceptive neuromuscular facilitation on patients with chronic ankle instability: A systematic review and meta-analysis," presents a comprehensive meta-analysis on the efficacy of Proprioceptive Neuromuscular Facilitation (PNF) in treating Chronic Ankle Instability (CAI). The study is well-structured and includes a thorough search of multiple databases, robust statistical analysis, and a clear presentation of results.

Abstract:

The abstract effectively summarizes the study's objectives, methods, results, and conclusions. However, it would benefit from a more detailed description of the specific outcomes measured (e.g., specific balance tests and muscle strength assessments).

Introduction:

The introduction provides a solid background on CAI, its prevalence, and the importance of rehabilitation. It appropriately introduces PNF as a potential intervention and sets the stage for the meta-analysis. The literature cited is relevant and up-to-date.

Methods:

The methodology section is detailed and transparent. Multiple databases for literature searches enhance the comprehensiveness of the review. The criteria for study inclusion are clear and appropriate. The use of STATA 12 for meta-analysis is standard and acceptable.

Specific Comments on Methods:

1. Literature Search: The range of databases searched is commendable. However, including the search terms used in each database for reproducibility might be beneficial.

2. Selection Criteria: The criteria for selecting studies are well-defined. Yet, a flowchart summarizing the selection process would improve clarity.

3. Statistical Analysis: Statistical tests and software are appropriate. The manuscript should briefly justify the selection of STATA 12 and the specific meta-analytic methods used.

Results:

The results section is comprehensive and clearly presented. The inclusion of both quantitative and qualitative analyses adds depth to the findings. The statistical significance and effect sizes are clearly reported.

Specific Comments on Results:

1. Balance Ability: The improvement in balance ability measured by YBT and SEBT is well-documented. Including figures or tables summarizing these results would enhance readability.

2. Muscle Strength: The results related to muscle strength improvements are significant. However, additional details on the specific tests used to measure muscle strength would be helpful.

3. Pain and Questionnaire Scores: The significant improvements in VAS and ankle instability questionnaire scores are notable. It would be beneficial to discuss the clinical relevance of these findings.

Discussion:

The discussion appropriately interprets the findings, highlighting the clinical implications of PNF in improving balance, muscle strength, and pain in CAI patients. The study's limitations are acknowledged, and suggestions for future research are provided.

Specific Comments on Discussion:

1. Interpretation of Results: The discussion effectively relates the results to the broader context of CAI treatment. However, it could benefit from a more in-depth comparison with existing literature.

2. Clinical Implications: The potential for PNF to be integrated into standard CAI rehabilitation protocols is well-argued. Further discussion on the practical implementation of PNF in clinical settings would be useful.

3. Limitations and Future Research: The limitations related to heterogeneity among included studies and potential publication bias are acknowledged. Suggestions for future research could be more specific, particularly regarding the types of studies needed to address current gaps.

Conclusion:

The conclusion succinctly summarizes the main findings and their implications. It aligns well with the objectives and results of the study.

Minor Comments:

1. Grammar and Style: The manuscript is generally well-written. However, a few minor grammatical errors and awkward phrasings should be addressed.

2. Formatting: Ensure consistency in formatting, particularly in headings and subheadings. Including figures and tables within the results section rather than at the end of the manuscript would improve flow.

Overall Recommendation:

The manuscript is a valuable contribution to the literature on CAI rehabilitation and the use of PNF. It is well-structured, methodologically sound, and clearly presented. With minor revisions to improve clarity and detail in certain sections, it is suitable for publication in PLOS ONE.

Suggested Actions:

1. Include a detailed flowchart of the study selection process in the methods section.

2. Provide more specific details on the measurement of outcomes, particularly muscle strength.

3. Enhance the discussion by comparing findings with existing literature and providing more specific suggestions for future research.

4. Address minor grammatical errors and improve formatting consistency

Reviewer #2: Congratulations to the authors for the study. This is a systematic review of chronic ankle instability. The data are solid and the results interesting, and the meta-analyses show promising results, justifying its publication.

6. PLOS authors have the option to publish the peer review history of their article (what does this mean?). If published, this will include your full peer review and any attached files.

Reviewer #1: **Yes: **Rahman Sheikhhoseini

Reviewer #2: No

---

## [Author Response · Author response to Decision Letter 0]

9 Sep 2024

Dear Reviewer 

We deeply appreciate your valuable comments concerning our manuscript. The suggestions are all very helpful for revision and improvement. We have carefully studied these comments and made related modifications, which we wish could meet with your approval.

The modified content has been marked yellow in the text.

1. Include a detailed flowchart of the study selection process in the methods section.

Thank you for this suggestion. Revisions have been made to the manuscript.

2.2 Literature Inclusion、Exclusion Criteria and Outcome Indicator

According to the PICOS principle, the inclusion、exclusion criteria and outcome indicator for the literature are as follows (See Table 1): 

Table 1. PICOS framework

Parameter Inclusion criteria Exclusion criteria

P (population) Adults (>18yrs) population from chronic ankle instability：

1) Subjects had at least one history of ankle sprain in the past 12 months, causing pain and swelling, and the time to lose normal function within 1 day or more

2) The affected ankle of subjects felt "soft leg", and/or repeatedly sprains and/or "un-stable" Non-chronic ankle instability population, ankle surgery history, and animal research

I (intervention) Proprioceptive neuromuscular facilitation(PNF), either alone or in conjunction with other measures Non-clinical trials and studies without intervention designs

C(comparison) Any alternative treatment includes standard care, placebo, or no treatment 

O (outcomes) Functional scales、Balance tests、visual analog scale、Muscle strength、Joint Position Sense、Range of Motion Studies from which data is not extracted or original data are inaccessible

S (Setting) Language of Publication: Articles published in Chinese or English. Non-Chinese/English documents; non-original research, such as reviews

Note: Functional scales including the Ankle Joint Functional Assessment Tool Questionnaire (AJFAI) and the Cumberland Ankle Instability Tool (CAIT); Balance tests include the Star Excursion Balance Test (SEBT) in directions: Anterior (ANT), Anterolateral (ALAT), Anteromedial (AMED), Medial (MED), Posterior (POST), Lateral (LAT), Posterolateral (PLAT), and Posteromedial (PMED), as well as the Y Balance Test (YBT); Visual Analog Scale for Pain (VAS); Muscle strength assessments covering relative peak torque during Plantar flexion, Dorsiflexion, Inversion, and Eversion; Inversion Joint Position Sense; and Ankle Dorsiflexion Range of Motion (DFROM).

2. Provide more specific details on the measurement of outcomes, particularly muscle strength.

Thank you for this suggestion. Revisions have been made to the manuscript.

3.4 Effect of PNF on Muscle Strength

Muscle strength was assessed using both an isokinetic dynamometer and an isometric handheld dynamometer. The former was used to measure the relative peak torque of the muscles [40], whereas the latter evaluated the maximum peak force [14]. Higher values of relative peak torque and maximum peak force indicate greater muscle strength. Since the units of measurement for the two methods differ, the standardized mean difference (SMD) was chosen for data synthesis.

3. Enhance the discussion by comparing findings with existing literature and providing more specific suggestions for future research.

Thank you for this suggestion. Revisions have been made to the manuscript.

In the second, third, fourth, and fifth paragraphs of the discussion, the comparison between the research results and the existing literature is added.

Second paragraph: The results suggest that PNF techniques can improve lower limb postural control, aligning with previous findings [23, 38]

 Third paragraph: Compared to other interventions, PNF training also involves additional sensory stimuli (tactile, visual, or verbal) to better facilitate neuromuscular responses and enhance the subject's neural control of the muscles[58].

Fourth paragraph: PNF can alleviate pain, help patients avoid discomfort during movement and daily activities, improve joint function and quality of life, and reduce the risk of secondary injuries.

Fifth paragraph: A meta-analysis conducted by AKASAKI H et al. [64] on the effects of PNF on joint position sense showed that PNF did not provide significant advantages, which is consistent with the results of this study.

In the article's conclusion, suggestions provided by future research are added.

In clinical treatment, PNF training is recommended for improving pain, balance ability, and muscle strength in patients with CAI. However, additional therapeutic approaches should be incorporated if patients with CAI experience significant mobility limitations. For future research directions, we propose the following: (1) To optimize research protocols and experimental designs by using PNF as the sole variable in the experimental group, and including other interventions, either without PNF or with sham PNF, in the control group to determine the combined effects of PNF and other interventions; (2) To increase the sample size and conduct medium- and long-term follow-ups on CAI patients treated with PNF to assess whether the improvements are sustained over time; (3) To further conduct high-quality randomized clinical trials in clinical settings to provide more evidence-based support for clinical application.

4. Address minor grammatical errors and improve formatting consistency

Thank you for this suggestion. Revisions have been made to the manuscript.

The manuscript has been checked and revised.

---

## [Editor Report · Decision Letter 1]

18 Sep 2024

Effect of proprioceptive neuromuscular facilitation on patients with chronic ankle instability: A systematic review and meta-analysis

PONE-D-24-25236R1

Dear Dr. Wang

We’re pleased to inform you that your manuscript has been judged scientifically suitable for publication and will be formally accepted for publication once it meets all outstanding technical requirements.

Kind regards,

Renato S. Melo, PhD

Academic Editor

PLOS ONE
---

## [Editor Report · Acceptance letter]

29 Dec 2024

PONE-D-24-25236R1 

PLOS ONE

Dear Dr. Wang, 

I'm pleased to inform you that your manuscript has been deemed suitable for publication in PLOS ONE. Congratulations! Your manuscript is now being handed over to our production team.

Kind regards, 

on behalf of

Dr. Renato S. Melo 

Academic Editor

PLOS ONE